# Pandemic Data Collection, Management, Analysis and Decision Support:
# A Large Urban University Retrospective

Namrata Banerji
banerji.8@osu.edu
The Ohio State University
Columbus, Ohio, USA

Steve Chang
schang@osc.edu
Ohio Supercomputer Center
Columbus, Ohio, USA

Andrew Perrault
perrault.17@osu.edu
The Ohio State University
Columbus, Ohio, USA

Tanya Y. Berger-Wolf
berger-wolf.1@osu.edu
The Ohio State University
Columbus, Ohio, USA

Mikkel Quam
quam.7@osu.edu
The Ohio State University
Columbus, Ohio, USA

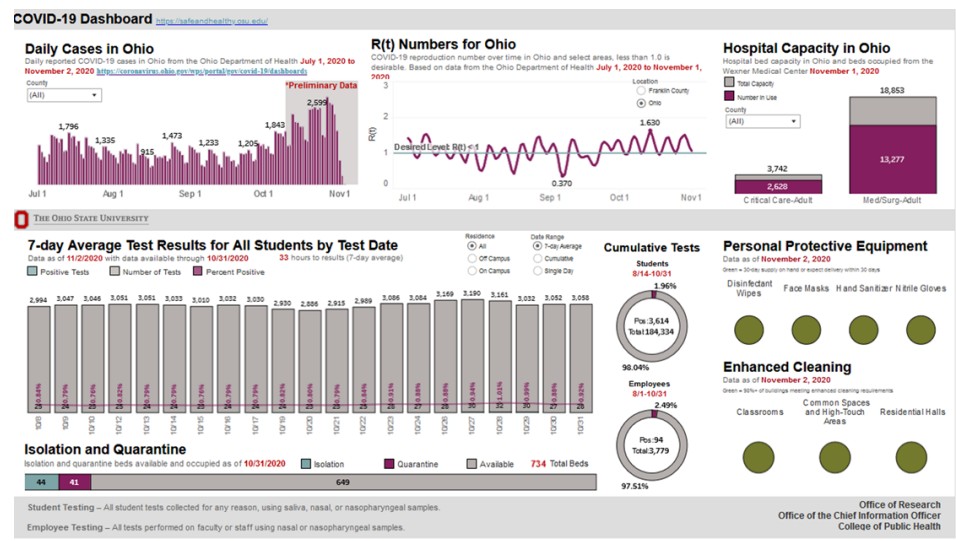

**Figure 1.** Archived OSU Safe & Healthy COVID-19 Dashboard for November 2, 2020

## Abstract

The COVID-19 pandemic has disrupted the world. During this crisis, data has emerged as a critical resource for understanding, monitoring, and mitigating the impact of the disease. We present The Ohio State University's data-driven framework for comprehensive monitoring of the COVID-19 pandemic. We discuss the challenges associated with data collection and investigate the roles and limitations of data analysis in supporting intervention choice and implementation strategies amid the complexities of the pandemic as it unfolded. Balancing privacy, consent, and transparency and ensuring the responsible handling of sensitive information is crucial in maintaining public trust. We examine privacy-preserving techniques, ethical frameworks, and legal regulations aimed at safeguarding individuals' rights while harnessing the power of data. In our experience, conscientious data architecture provided a foundation for meaningful ethical applications of data products, which not only helped mitigate the current crisis, but also can provide valuable insights for better addressing future public health emergencies.

*CCS Concepts:* • **Information systems** → **Database administration**; • **Applied computing** → *Health care information systems.*

*Keywords:* datasets, public health, data management, ethics

**ACM Reference Format:**
Namrata Banerji, Steve Chang, Andrew Perrault, Tanya Y. Berger-Wolf, and Mikkel Quam. 2023. Pandemic Data Collection, Management, Analysis and Decision Support: A Large Urban University Retrospective. In *epiDAMIK 2023: 6th epiDAMIK ACM SIGKDD International Workshop on Epidemiology meets Data Mining and Knowledge Discovery, August 7, 2023, Long Beach, CA, USA.* ACM, New York, NY, USA, 8 pages.

## 1 Introduction

The onset of the COVID-19 pandemic in early 2020 was one of the most significant and life changing events for everyone on the planet, impacting everything from small businesses to entire countries. In case of educational institutions, the indefinite suspension of classes, upending of every traditional aspect of academic and student life, and the transition to virtual education was stressful for students, staff, and faculty alike. The Ohio State University (OSU), a large urban educational institution, undertook a massive policy response to support the continuing function of the university by monitoring and managing the dynamics of the pandemic on and around its campuses. Putting together a coalition of epidemiologists, data scientists, public health policy makers was only the first step of what shaped up to be at least a three year marathon. Data was at the center of the whole process, both as the decision enabler and as the product of many of the contributing efforts. To make data actionable required the work of many teams and several iterations of cleaning, analysis and inference, and visualization. In this paper, we present the overall data-focused aspects of the process, highlighting the achievements and the hindrances, as well as the major takeaways, so that we are better prepared for future public health emergencies or other large scale collective responses. This manuscript, besides serving as a piece of institutional memory, communicates in detail the various obstacles encountered in the handling of the mammoth data for the data science community to be aware of. Among the main takeaways we consider the effectiveness of the data driven approaches for managing the pandemic response, the need for an institutional data infrastructure, and the importance of a well organized team of experts and professionals working together towards a well-defined goal.

## 2 Overview

The Ohio State University stood up the Comprehensive Monitoring Team (CMT) [4] to include a framework of support for data driven decisions for pandemic management, including robust case finding (via serial mass administration of individual PCR tests with rapid in-house processing), locally administered isolation of cases, contact tracing and quarantine of close contacts, as well as data integration, analysis, modelling, risk evaluation, policy recommendations, and intervention implementation based upon knowledge derived from individual case management, subsequent viral

(genomic) sequencing, large scale syndromic surveillance and evidence of environmental (wastewater and dust) shedding [6, 12, 14, 15]. Here we present the core of the data component of this system that integrated data from various testing centers, conducted daily analyses, and represented data in formats usable by the leadership to support both individual level contact tracing and the university's policy response to the public health emergency. In the coming sections, we discuss the goal of setting up such a system, the implementation pipeline, data sources and some of the challenges and takeaways.

## 3 Goals

Building and maintaining such a huge framework and employing a whole workforce including faculty, students, healthcare workers consumes university resources at a large scale. The goals were the result of several rapid iterations of convergent conversations between the university administration and members of the CMT, as well as the consultations with external experts. The specific aims of the data components of the framework were as follows:

- **Tracking the positivity rate.** Positivity rate or testing positivity rate, defined as the percentage of tests reported that are positive [10], emerged early in the pandemic as the agreed upon indicator of the state of the population and the basis for comparing different populations [9]. We used the positivity rate, throughout the monitoring process due to a number of reasons, one of them being that this percentage (sometimes a fraction) was the most expressive and conveyed a more complete story than other measures such as absolute number of positive cases. It is true that 100% of the university population was not being tested, because there were exemptions (medical and otherwise) and non-compliants, but we had the data necessary to determine exactly what fraction of the population was being tested. This was the best metric that we could monitor from the data and information available to us at the time, and it never became a cause for concern.

- **Contact tracing.** Removal of positive and potentially positive cases from the population is key for suppressing the spread of the virus [8, 17]. It was necessary to provide contact information for people who tested positive and to identify and contact their close contacts in order to isolate and quarantine them, respectively.

- **Understanding the micro trends and risks based on events.** To understand the dynamics, the risks, and the implications of the pandemic for various subpopulations it was necessary to provide the ability to zoom in on specific time intervals and subgroups in the data. Examples of the questions asked include: How does fall break or Halloween behaviour change/impact infection rates? Is there an increased risk of students in a 4-person suite over a 2-person

dorm room? How do the risks associated with in-person classes compare with hybrid or remote classes?

- **Supporting daily policy decisions of a large urban university.** Daily decisions supported by data included the choice of a testing strategy and protocol, transition to hybrid vs online only classes, occupancy in classrooms, vaccination and masking requirements, etc. Having access to the right data was essential. The testing protocol [3, 16] was more strict in the early days of the pandemic, requiring all students who live in residence halls or who have at least one in-person class to test at least once every week. The requirements were relaxed in the subsequent semesters. Testing mandates were also in place around holidays, for example, students were required to test before a Thanksgiving break and after. The WiFi data was often utilized to get a sense of how many students were still residing in the dorms over the break, and how many went home.

- **Reducing burden in the wider population.** OSU Columbus campus is a large urban campus with highly permeable boundary in the center of a city. In order to contain the pandemic, the infection rates needed to be controlled both on and around campus. Moreover, the university sought to mitigate the export of infections to communities beyond its campuses. College students mix with the city population and visit their family over academic breaks, potentially increasing the risk of transmission to vulnerable community members. Recommending and at times requiring testing before the academic breaks was one such measure taken to reduce the burden on vulnerable immuno-compromised population outside the university.

## 4 Implementation

OSU has 68,000 students, 12,000 of which reside in residence halls during a regular year. During the pandemic, about 8,000 students were in residence halls and were required to test weekly. Additional students, faculty, and staff were testing voluntarily. At its peak, more than 30,000 tests per week were processed.

Multiple teams across Information Technology support, Student Life, Translational Data Analytics Institute (TDAI), Infectious Disease Institute (IDI), University Medical Centers, College of Public Health, and many more were responsible for standing up a system that would be in place for at least the next 3 years. The data environment was a secure and flexible environment that allowed for dynamic data definition and integration of data from at least 56 sources when it was introduced. (The number of data sources grew to over 100 by the end of 2022.) Initial data sources included testing data together with the administrative data of student information, residence and permanent addresses, demographics, class registration, residence layout, class and college affiliations, WiFi

access point information, and much more. The pipeline is illustrated in Figure 2 and is described very briefly below.

- Primary test data was transmitted into the internal secure data environment via electronic file transfer multiple times a day.
- Additional attributions from other internal OSU systems (Identity management (IDM), Student Information Systems (SIS), Student Life, etc.) were preloaded and updated according to the system's change protocol (e.g. each semester).
- Test results and internal data were combined into a cohesive reusable dataset (AKA the "gold table").
- Analysts and dashboard builders utilized a common source for all reports and visualizations.
- Data was also sent to Helpspot/Salesforce to support case investigation and contact tracing efforts.

### 4.1 Data description and daily analysis

Among the 50+ tables and views that were maintained on AWS, there were 10-12 datasets, described below, that were most frequently accessed for daily analysis reports.

- **'Gold' dataset of people**: This view is derived from multiple tables, that contain individuals' unique identifiers, demographic information such as gender, race, ethnicity, age, home and campus address, affiliation with the university, affiliation with an OSU campus, indicators of whether their on or off campus, student housing residence, etc. There are roughly 2.5 million entries in this dataset, with updates at regular time intervals of changing affiliations, addresses, and other variables.

- **'Gold' dataset of tests**: Similar to the gold person table, this is also a derived view of data on tests administered by the university that combines variables like test provider name, test administered time, test result time, test result, type of test conducted, etc. It also contained some of the demographic information and addresses so that quick results could be obtained by running simple queries, without joining multiple tables.

- **Dataset on off campus residence housing**: This dataset contains information on what organizations individuals are a member of, whether they are an active member, whether they live in the organization housing, etc. This was a particularly useful dataset at the beginning of the pandemic as many outbreaks occurred in off-campus residence houses, which were analyzed for patterns [13].

- **Dataset on contact tracing**: Each actionable positive test result generated a ticket, which is entered into a SalesForce(TM) dataset of tickets. The metadata associated with each ticket included a unique ticket identifier, the person whose close contact this is, the person who is the close contact, both their information, the time and result of the test, whether that person had symptoms, whether that person is an OSU affiliate, etc. This dataset was important throughout the pandemic, since these tests and contacts were the focus of most of the analyses. Also, this dataset contained

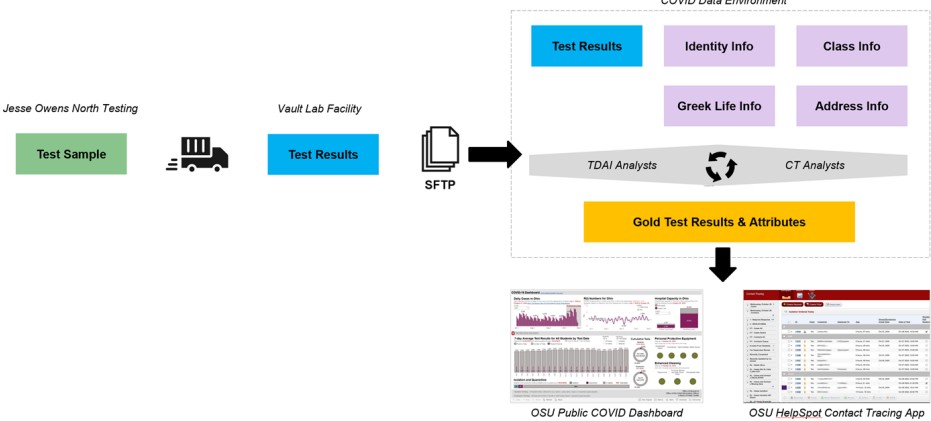

**Figure 2.** Data flow in the OSU COVID-19 monitoring pipeline.

data on positive tests even if they were not present in the gold test data table. This is because while the gold table only recorded tests that were administered by the university, the SalesForce(TM) tickets datasets contained information on other tests, some outside the university, as long as they were positive. This dataset was thus a good source for absolute number of positives in the university community, but not very good for computing rates, due to the absence of a denominator.

- **Datasets on class enrollment**: When the university re-opened for the Fall after the summer of 2020, a lot of classes were online, some were hybrid, and few were in-person. It was important to know if there was additional risk of infection for students enrolled in classes conducted in person, and decisions had to be made to combat the risk and spread of infections. The class enrollment datasets were key in this effort.

- **Datasets on vaccination**: Two datasets were maintained that contained vaccination information, one for students and one for employees (including staff). Although containing the same information in essence, the two were structured differently. The tables for students contained two date variables, one denoting the date of dose received, and the other indicating the date when the individual becomes fully vaccinated according to CDC guidelines. It also had variables corresponding to whether the individual had a vaccination exemption, whether the dose was CDC approved, the CDC code (e.g, 208 for Pfizer) [2], whether the shot was a booster, etc. On the other hand, the employee vaccination table contained columns on first vaccination date, second vaccination date, up to seventh vaccination date and the provider information for each in addition to the exemption and booster indications. Thus, the data analysis needed to produce the same results from the two tables needed to be different.

The initial daily analysis included breakdown of test positivity rate in each of the residence halls, between demographics, majors, and campuses. This was for internal consumption, pattern identification, and insight derivation. Much of this data and the derived analysis was private and was not made public. The results that did make it to the dashboard [3], as shown in Figure 1, were the aggregate and summary numbers on reproduction number, which is a standard epidemiological metric [7], the daily number of cases, the 7-day average, etc.[1]. Identification of close contacts of students residing in dorms was a large part of the daily analysis and the gold datasets were utilized to that end to produce a list of roommates and suitemates. A concise description of the analysis performed was first published in an initial report [4] in October 2020 and updated in a second report[5] in March 2021 by the CMT.

## 5   Challenges

The novelty, scale, and duration of the recent and ongoing pandemic were major challenges. Data collection, management, and analysis pipelines at this scale had no modern precedent and had to be designed as they were beginning to be used. Moreover, the timelines were drastically compressed and the requirements initially were changing frequently. In addition, some areas, such as close contacts or attendance of events, lacked data collection, and some critical data streams, including off-campus testing, were initially completely absent. Further, as most teams around the world, we initially lacked the full understanding of how to translate the questions into data and how to prioritize the variables and the analysis for decision support, particularly in the context of human behavior. Below are some of the issues that posed significant challenges to the team.

---

[1]The dashboard was awarded the A+ rating and selected as the best COVID-19 university dashboard by the "We Rate Covid Dashboards" panel of academics [1]

## 5.1 Data cleaning

The data was collected from numerous sources, some of which were manual entries and consequently had unavoidable human error. For example, a table of people in the database had the OSU unique identification (name.#) as the primary key, and the table of test results was supposed to have the same as foreign key. Typographical errors or null values in this identifier column resulted in our inability to correspond a test to an individual, causing a non negligible shift in the summary statistics. Once the problem had been identified, there was joint effort to clean it up, combining more than four data streams and reducing the number of unidentified tests to a number that would not change the inference. Yet, there were still a few individually unidentifiable entries in the datasets, albeit not a high number to raise a concern. Minimizing manual entry to data sources can reduce such issues by a considerable amount.

A similar problem was found in the table for employee vaccination records, with clearly wrong dates of doses. While most were due to errors, in some cases, employees were actually part of vaccination trials and had received a dose before any vaccination received emergency use authorization or approval for distribution to the general public. These cases were indistinguishable from the erroneous cases without careful manual investigation and knowledge of the regulatory frameworks and timing of numerous vaccine candidates from all over the world.

One of the challenges that the team immediately encountered while using demographic data was that there were a number of similar datasets, curated by different organizations at OSU, and used for different operational purposes. Re-purposing these for COVID-19 demographics analysis required that specific datasets and methodologies were employed for consistency. Part of the Human Infrastructure that was critical here were experts of the use of these legacy datasets to be able to share what nuances may have been encoded in the data, and to help determine the least wrong datasets and methods to use. This investigation eventually led to the creation of the "gold" datasets, which were so named because they were the COVID project's Gold Standard demographic associated with an individual or test.

These examples illustrate the need for expert data curation, close scrutiny of analysis outputs that consumed these data sources, efforts to minimize manual data entry, and for close collaboration with domain experts at every step.

## 5.2 Data storage, backup, documentation, and recovery

The volume of data generated by testing mandates as well as voluntary testing required careful consideration of large, yet quickly accessible and continuously backed up data storage. The ability to look up prior data was critical to understanding trends and the dynamics of trends, as well as comparing the outcomes of various past decisions. For continuously changing data, such as the daily updated test data, it is needed to maintain regular snapshots, checkpoints, and versions. This aspect was not fully appreciated initially and required significant efforts to redesign data architecture. We maintained two 'gold' datasets, one corresponding to people and demographics and one corresponding to tests' metadata. These derived datasets were cleaned and organized to our standards that would be the basis of further analysis. This cut down on the work of individual analysts so that those cleaning/organization steps would not need to be repeated. The 'gold' data of people, consisting of faculty, staff, students, and everyone else affiliated in some way with the university, updates significantly every semester overwriting previous data in the database (S3 environment). We would save a snapshot of the data every semester, but unfortunately initially the snapshots were taken towards the end of the semesters when students had already started leaving the campus. As a result of this, recently when we wanted to get a time series of positivity rates in residence halls, it was different from the original since we do not have the correct denominator. Recovering this information is possible, but requires integration of other data sources, demanding significant investment of resources, effort, and time. Majority of the people who were part of the university supporting the CMT and were responsible for setting up the system are no longer working at OSU. Moreover, early in the reopening of the university, the primary focus was on managing the pandemic and bringing down the positivity rate, and detailed documentation was not prioritized.

Mid semester migration from one homegrown case data management solution to an outside vendor was a major issue that required major investment and retraining and we are continuing to deal with this today from a data and analysis perspective. Roughly from August 2020 to November 2020, we had our positive test (case) data ingested and case investigation/contact tracing notes stored in a secured instance of a HelpSpot database integrating in some instances with REDCap surveys and pushing out to several communication platforms, but later we shifted to a Salesforce Health Cloud build, which assisted with future testing data variations, vaccine information, as well as some automatic reminder communications. The data had been migrated from the old table to the new one in theory, but in part user generated heterogeneity, as well as version control issues in the HelpSpot source data meant there continued to be gaps in the data ingested by Health Cloud (Salesforce) which do not have simple workarounds for analysis of all variables. We maintain several tables for the test information storage, but there are inconsistencies across those tables. More than one tables exist mainly because we derived simpler versions of tables with many columns that are not relevant for day-to-day analysis. One of the (intermediate) mother tables recently had one of its very important columns (the test specimen collection

time/date column) dropped from an integration during an update, and it should have been okay to just look it up in a derived or other related testing table had there not been major differences in the number of entries in the others.

The IT organization at OSU, then known as the Office of the CIO (OCIO) had embarked on a project prior to the COVID epidemic to move OSU Enterprise data off premises and onto Amazon Web Services (AWS). AWS was the obvious choice as the data storage platform, as much of the data were already present on the platform, and tools such as Amazon Athena were able to provide a layer of data abstraction so that disparate datasets could be queried in a consistent manner. That OCIO project to house these data in a consistent manner was fortunate; it would otherwise have added an additional layer of processing to export and synthesize data from various legacy systems. The other major consideration is that there are significant costs of using a commercial cloud service. While these were covered in part by the OCIO project, additional data storage for COVID data and the use of AWS tools such as Athena were incurred by the COVID project.

### 5.3   Data governance and ethical considerations

The university has a complex set of data governance regulations as do individuals' private health information, whether used in the healthcare or public health applications. While special authorization was granted to use some of the data in the pandemic emergency, security and privacy remained strict requirements. Each team member had training in handling secure and private data.

In addition to the standard data governance issues, dealing with the high resolution personal data has its own set of ethical issues. Ultimately, the main question was: what is the benefit of using a particular data source or performing a particular analysis and would it change the decisions or the pandemic dynamics? If so, was it necessary to use individual and identifiable data for decision making or could aggregate or coded information have similar utility? For example, while it is within the rights of the university to use the WiFi access point information to "follow" an individual or to understand who is within the same room, such information has a high 'icky factor' and should be used sparingly. Moreover, while initially it seemed that WiFi data would provide a good proxy for contact tracing, it turned out that the resolution of the data did not correspond well to the physical definitions of a contact. Ultimately, it was decided to use WiFi data in aggregate to assess population movements rather than individuals' proximity to other individuals. For example, WiFi data was used to estimate the number of students leaving campus over the weekend or the number of students present in an "in person" classroom. Moreover, the aggregate trends proved to be much more robust than the individual-based analysis and were significantly less time consuming. Additionally, adherence to the current applicable statutory guidelines for case

investigation, subsequent case management, and/or contact tracing may require some variation depending upon individuals' occupation, travel history, personal risk factors, immunocompetence, vaccination status, which could include certain specific preexisting conditions, medications, clinical care received, viral (variant/sub-variant) lineage, and/or disease severity. However, specific individuals' health information related to their experience with COVID-19 would largely not meaningfully determine macro-level prevention policy or interventions in the university context independently from aggregate trends and information in the wider public health policy guidance, which are separately informed by individuals' public health, laboratory testing and clinical health records. Therefore, particularly those sensitive individual level data, especially health data were collected and subsequently shared only to the extent they would have 'meaningful use', within the data user groups' spheres of control, stated goals, and purview (i.e. healthcare providers would have access to information relevant for managing patient care; public health authorities would have access to information relevant to determining specific application of disease management protocols for individuals and/or groups; occupation health, workplace, and student life safety personnel would have limited access to information relevant to adherence with applicable disease prevention laws and policies aimed at risk reduction, such as adherence to testing, vaccination, and isolation/ quarantine requirements in some instances).

## 6   Takeaways

### 6.1   Behavior over analytics

The main takeaway of our data-supported pandemic monitoring framework is the same as the main takeaway for dealing with the COVID-19 pandemic world-wide: ultimately, the main determinant of the success of the system hinges on modifiable human behavior, rather than the sophistication of the analysis. No improvement in the accuracy of the analysis of the effect of masking in a given setting (i.e. library, classroom, laboratory, or healthcare setting) is meaningful if people would not (continue to) comply with an indoor mask mandate. Similar limitations became apparent with both pharmaceutical and non-pharmaceutical interventions, even as evidence increasingly substantiated benefits and new sub-variants emerged, populations' apparent risk tolerance grew and spread.

### 6.2   Communication is key

Working with a team this large, with people from vastly diverse backgrounds, communication between the teams becomes an essential component. A major part of the analysis was being carried out by graduate student employees, who were sometimes not aware of things like floor structure in dorms, testing protocols, vaccination mandates, etc.,

which were important analysis components. Similarly, the modelling team was involved in building risk models, models for testing strategy development, etc. that relied on domain knowledge outside of mathematics or computer science. Clearly, experts in every relevant domain (epidemiology, public health, student residence life, university logistics and operations, etc.) need to be constant partners in the analysis.

### 6.3 Equity Considerations and singling out demographic groups

When patterns appear to be emerging within specific groups or sub-demographic, there may be an equity oriented opportunity for targeting or strengthening an intervention but there may also be a bias in the observed signal. One group may in fact be more often in situations involving exposure to infectious persons, or engaged in more risky behavior than others, as we occasionally discovered from data analysis. However, available policy level changes may not have been feasible solutions and were not always ultimately enacted. What we started to see in the data raised questions on ethics and trustworthiness of data enabled interventions, without context or corroboration. Some solutions aimed to address one groups perceived or real deficiency in access to resources or excessive exposure could foster stigma, or loss of other resources in unanticipated ways. After careful consideration, it was agreed that singling out a group was often not enough of a value addition or could do more harm than good. In some cases, trends observed initially in one population or group were indicative of larger trends that could be addressed by policy shifts relevant to the whole community, which would address both the observed inequity and mitigate for known unintended consequences.

### 6.4 Micropatterns significant, but not usable in hindsight

The reflections on the decisions made over the course of three years showed that the micropatterns and the microtrends observed in the data had little to no effect on those decisions. Observations that a certain subgroup engaged in activities that increased the risk of the spread of the infection did not prompt the authorities to take measures to shut down those activities in many cases because it was either not cost effective or unethical to do so. These data nuances did provide information but it was not actionable. In retrospect, however, the information's main utility was in the fact that no single critical subgroup was the key to the solution. The scale of the phenomena did not lend itself to a single pathway of solution or a single target group. Patterns that we learned in settings like an early long term care facility were also observed later in dorms, sorority and fraternity houses and athletics teams and they led to better population level responses. A good example would be the limitations of certain kinds of tests for transmission suppression. The Big10 testing program involved daily testing of athletes during their competition

season, given that team members were often unable to mask and physically distance in some sports. Unfortunately, when transmission started to increase rapidly in late autumn 2020 as sports teams re-started their compressed seasons, even daily testing with rapid results was insufficient to suppress transmission, largely because the particular test used did not detect all infectious individuals immediately. By the time one tests positive on an antigen test, like those in use at that time, a person may have already been infected and infectious for a few days, meaning potentially exposing others and continuing transmission chains. Antigen tests are useful for rapid diagnosis particularly when symptomatic but are not always ideally suited for early enough detection to reduce spread in a serial testing model. OSU opted for developing and deploying swift, minimally invasive (saliva based), highly specific, highly sensitive, PCR testing, shown to be able to detect pre-symptomatic and asymptomatic infections (eventually even processing results with its own PCR testing and sequencing lab capable of thousands of tests per day). Although they were not as fast as antigen tests, the average turnaround time was less than 24 hours during much of the semesters' most populated period. This was a scenario where tracking a micropattern in a particular well-observed and well-resourced group gave us really good information of what and how we should be optimizing testing resources and working within their limitations with the larger university community's population.

### 6.5 Data infrastructure

The overall data infrastructure consists of cyberinfrastructure (compute, storage, networking, cloud and web services), information infrastructure (data and metadata management, search, archiving, cataloging, and digital services), and analytics infrastructure (data integration, harmonization, and analysis). The large volume of data collected, collection rate, distributed team setting, potential errors, inconsistencies, and variations in reporting standards, and changing objectives all strained and challenged existing data infrastructure at OSU and necessitated expansion of that infrastructure. Moreover, COVID-19 management provided a great case-study and emphasis on the fact that data infrastructure integrates cyber-, information, and data services infrastructures through **human infrastructure**. Building the human infrastructure is both the most critical aspect and the hardest to implement of any data infrastructure. We have seen personnel migrate out of the team, and the university, and when that happens, they take institutional knowledge with them. Replacing personnel in such a fast paced environment entails a lot of rigorous training that newer team members have to go through within a very short period of time. Even after being on board, it takes significant time to bring them up to speed, which often creates a bottleneck.

### 6.6 Scale

The sheer volume of COVID-19 data generated from testing and vaccination overwhelmed existing data management systems of the university as well as the state. Scaling up data infrastructure and analytical capabilities to handle large-scale data collection and analysis proved to be a significant challenge, but one that can definitely be overcome.

## 7 Comparison between similar systems in place nationwide

The COVID-19 pandemic was monitored worldwide, and any attempt to track rates or contain the outbreaks had to involve systems governing huge amounts of data. Among the humongous number of research papers out there utilizing the pandemic data, very few of them talk about the nuances of the data collection and storage mechanisms deployed. For example, a paper [18] from University of Michigan talks about collecting environmental surveillance data in order to estimate infection risk. This direction of research and analysis was popular in a lot of organizations and was a good means of estimating risk of infection within the campus from sources like dust and sewage water, including OSU [6, 14, 15]. Another paper [11] discusses digital health research and tracking in general, but in the light of the pandemic and how it impacted practices. Their concerns are very similar to ours, but unlike their generic view, we provide a complete story of a real experience with a series of issues faced and tackled in an urban institute.

## 8 Conclusion

We hope that the COVID-19 pandemic was a one-off unique event, never to be repeated. Yet, we should be prepared to respond to a similar event by learning from our experience. We hope that the OSU CMT work presented here can serve not only as a blueprint, but as a guide for considerations, priorities, and potential pitfalls, should the response at this scale be ever needed.

## Acknowledgments

We would like to acknowledge the work of many people who have contributed to the effort of enabling the data driven approach to monitoring and managing the COVID-19 pandemic at the Ohio State University: the entire Comprehensive Monitoring Team (CMT), Case Investigation and Contact Tracing Team, CMT student analysts, CMT/IDI Modeling Team, Applied Microbiology Services Lab, Testing Operations Team, Student Life Isolation and Quarantine Team, Student Health Services, Employee Health Services, local and state public health authorities, dashboard developers, and the OTDI team, including D&A data engineers, data governance team, network administrators, and enterprise security.

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
