# OpenReview forum: "Pandemic Data Collection, Management, Analysis and Decision Support: A Large Urban University Retrospective"
_KDD.org/2023/Workshop/epiDAMIK — KDD 2023 Workshop epiDAMIK_

### Official Review · Reviewer_oZ8Y · 2023-06-16
**Good experience sharing but not a well-formed research paper**

**Rating:** 1
**Confidence:** 3

**Review:**

Summary Of The Review:

This paper discusses the role of data in managing the COVID-19 pandemic, focusing on its collection, management, analysis, and application in decision-making. The authors present The Ohio State University's data-driven framework for monitoring the pandemic, including data sources, methods, and technologies used for case finding, contact tracing, and visualization. They discuss challenges such as privacy concerns, data quality, and the need for harmonization across different sources. The paper also explores ethical considerations in data usage during the pandemic. The authors highlight the importance of data architecture, teamwork, and ethical frameworks in addressing public health emergencies. The paper concludes with key takeaways and lessons learned for future public health emergencies.

Pros:
1. The authors possess extensive knowledge and experience in managing COVID-19 at The Ohio State University, providing valuable insights and practical examples that can benefit other systems.

2. The paper offers a comprehensive discussion of the university's policies.

Cons:
1. Lack of references and comparisons

The paper lacks citations to substantiate and compare the findings and approaches presented, which is a notable deficiency. For instance, it is essential to clarify the definition of the "positivity rate" in this paper and provide a rationale for its use, which would benefit from external support. Additionally, the paper displays "R(t) Numbers for Ohio" in Figure 1, but fails to mention or discuss this important metric in the text, warranting a comprehensive review of the relevant literature to enhance its explanation. Similarly, the utilization of terms like "gold table" and "'gold' data of people" could be less perplexing if supported by appropriate references.

Furthermore, considering the abundance of existing pandemic surveillance systems, it would be advantageous to examine their operational mechanisms. This examination would enable the identification and comparison of the strengths and weaknesses of the system presented in the paper.

2. Ambiguous statements

Several statements in the paper lack clarity and precision, leading to confusion among readers. Additionally, the paper fails to provide a comprehensive summary of the data entries presented in the tables, leaving readers unsure about the specific information contained within them. Moreover, the paper lacks explicit descriptions of tasks, analyses, or well-defined evaluations, despite drawing conclusions and using phrases such as “No improvement in the accuracy of the analysis of the effect of masking in a given setting” and “After careful consideration, it was agreed that singling out a group was often not enough of a value addition or could do more harm than good.”

Here are additional instances of imprecise terminology lacking explicit definitions or thorough evaluations of their scope.

Section 5.1:

“Typographical errors or null values in this identifier column resulted in a non negligible shift in **the summary statistics**, given the **enormous number** of tests conducted. Once the problem had been identified, there was **joint effort to clean it up**, combining **more than four** data streams and reducing the number of unidentified tests to **a number** that would not change **the inference**. Yet, there were still **a few** individually unidentifiable entries in the datasets, albeit **not enough a number** to raise a concern. Minimizing manual entry to data sources can reduce such issues by **a considerable amount**.”

Section 5.2:

“The data had been migrated from the old table to the new one in theory, but in part **user generated heterogeneity**, as well as version control issues in the HelpSpot source data meant there continued to be **gaps** in the data ingested by Health Cloud (Salesforce) which do not have simple workarounds for analysis of all variables. We maintain **several tables** for the test information storage, but there are **inconsistencies** across those tables. More than one tables exist mainly because we derived simpler versions of tables with many columns that are not relevant for **day-to-day analysis**.”

Section 7.2

“**One group** may in fact be more often in situations involving exposure to infectious persons, or engaged in more risky behavior than others, as we **occasionally** discovered from data analysis. However, available policy level changes **may not have been feasible solutions** and were not always ultimately enacted.”

3. Unaddressed privacy concerns

The authors argued that they would “examine privacy-preserving techniques” and “security and privacy remained strict requirements”. While in section 7.2, the author also said “while it is within the rights of the university to use the WiFi access point information to “follow" an individual or to understand who is within the same room, such information has a high ’icky factor’ and should be used sparingly.” Despite this, “it was decided to use WiFi data in aggregate to assess population movements rather than individuals’ proximity to other individuals”. Furthermore, the data is “shared”; “health data were collected and subsequently shared only to the extent they would have ’meaningful use”. It would be useful to clarify who it was shared to, what was shared, what training team members had, and describe in more detail the type of data that is collected and disseminated from tracking student’s WiFi locations, seemingly without their knowledge or permission.

4. Other minor problems

Figure 1 in the paper was presented without any accompanying explanations, leading to confusion among readers. The lack of clarification makes it difficult to comprehend the purpose and significance of the "Personal Protective Equipment" and "Enhanced Cleaning" sections, both of which are represented by equal green circles in the figure.

Also, “Behavior over analytics“ should be section 6.1 rather than section 7.

In general, the paper offers valuable experience in data management during the COVID-19 pandemic. However, there are several areas that require improvement. First, the paper should include more references and comparisons to support its findings and approaches. Additionally, the analysis section would benefit from a more detailed explanation of the methodologies employed. The clarity and logical presentation of results and takeaways also need to be enhanced. Furthermore, the paper should address privacy concerns associated with the data management practices discussed.

---

### Official Review · Reviewer_8BXR · 2023-06-29
**Good work for data monitoring and collection**

**Rating:** 4
**Confidence:** 5

**Review:**

The paper presents a specific diagram of COVID-19 data tracking, monitoring, and collection. The work is practically meaningful and valuable to various communities for future studies:

1. The paper presents a clear and comprehensive process from collecting test samples to final dashboard exhibitions, which provides valuable paradigm experience for data collecting and processing, particularly for college and education communities.

2. The collected data are valuable for public policy and AI modeling communities. For instance, the work uses Wifi data for individual monitoring and contact tracing, which may help establish contact networks and provide a better understanding of how disease can spread within schools.

Despite the meanings of the data collection process, we wish to understand more about the collected data. For instance, it would be good to include non-private or non-sensitive statistical analysis and visualization of the data for the presentation or the final paper.

---

### Official Review · Reviewer_HX1g · 2023-06-30
**OSU Covid-19 Data Retrospective**

**Rating:** 2
**Confidence:** 3

**Review:**

# Quality
The paper is well written and provides a comprehensive retrospective of OSU's pandemic response

# Clarity
The paper seems to offer examples rather than comprehensive descriptions of data. Understandably difficult to cover everything, but in a retrospective like this, comprehensive analysis is going to be more useful.


# Originality
Similar pandemic response retrospectives exist for other institutions, while interesting seeing OSU's work, originality is low.


# Significance
While a good snapshot of the work that occurred at a large scale public university, I feel the lack of originality reduces the overall significance.


The paper offers unique and detailed insight into the Ohio State pandemic response process and data collection, detailing the successes and failures of different applications and the lessons that the university leadership learned in the application of these policies. The lessons detailed would be applicable to another pandemic situation should one arise, making iterations on this faster and producing more useful insights more quickly.

The paper itself, while interesting to read and learn from, lacks large unique insights, rather agreeing with many other retrospectives with minor shifts in lessons and policies.

NOTE: it looks like sections 6/7 are incorrectly labeled (section 7 uses a \section{} tag rather than a \subsection{} tag)

---

### Official Review · Reviewer_TzBF · 2023-06-30
**Review of Data Collection, Management, Analysis and Decision Support During COVID-19: A Retrospective from The Ohio State University**

**Rating:** 3
**Confidence:** 3

**Review:**

Summary:
This paper discusses the large undertaking of collecting, processing, and reporting COVID-19 data from the Ohio State University. This paper makes note of the challenges and missteps faced in data processing, and the lessons learned from this experience during the pandemic.


Clarity: This paper was clear and easy to follow.

To improve upon the clarity, I would suggest the following:

--Ensure that the “aims” listed are discussed in later sections in the paper. The first aim of “tracking the positivity rate” is not mentioned at any other point in the paper. It is not clear from this paragraph which positivity rate is being tracked (university affiliates?), and whether any weighting scheme was applied to the data. Similarly, the second aim is “contact tracing” however there is no further discussion of how contact tracing was done and/or recorded. It is unclear whether this is really an aim of the data component, or whether this is considered too far downstream. If both of these were aspects of the data framework, they should be expanded on in the implementation section.

--In the figure 2 schematic, it would be helpful to highlight the data processing / management steps or programs used to convert from the gold test results to the dashboard and contact tracing app. Additional details could be added to this figure.

--The term, “human infrastructure” is bolded in section 7.4, yet this term is not defined. It would be beneficial to define what this term means in the context of this paper, as this term may not be familiar to many readers.

Minor comments on clarity:


--In the “implementation section” and in figure 2, a number of abbreviations are used that are never spelled out. Writing out these abbreviations would clarify the paper and the data schematic (e.g., IDM, SIS, STFP, TDAI).


--The discussion surrounding issues with salesforce data is unclear. The authors mention “user generated heterogeneity” and “version control issues”, however the link between those issues and what is causing gaps in data is not fully apparent.

--Figure 1 is not mentioned at all in the paper. It would be useful to include a discussion of who is using / viewing the dashboard and how frequently it was used.  That would give an indication of how the data was being used by the community / decision makers at OSU.



Originality:  The work is original in that it is the only paper to describe the data-driven processes occurring at the Ohio State University. However, many of the points made are not unique, and seem to highlight issues with this data management system. Lessons such as the need to “minimize manual data entry”, work with experts in “every relevant domain”, and following ethical guidelines regarding data privacy are not ideas original to this project. To highlight the originality of this work, it would be helpful to have a small review of literature section that discusses how this project improves or differs from similar undertakings at large universities.



Significance: The significance of this paper could be improved by including more actionable messages to future data systems and teams. Significance could also be improved by noting how this data was used for decision making. One of the goals listed in section 3 is to “support daily policy decisions”. However, throughout the paper there is little indication of how the data that has been acquired, processed and presented informs decision making. Including additional examples of how this data was used would be very beneficial. The significance would also be boosted by discussing how individual COVID-19 testing data was integrated (if at all) with wastewater data and/or genomic data to inform university policy.



Pros:

--Well written paper

--Concisely and clearly presents aims of a data-driven framework

--Clearly explains many of the pitfalls that can occur in data management, and acknowledges that during the pandemic, some best-practices (such as recording all steps along the way) were not followed due to the need to provide numbers to decision makers.

-- Provides nice examples of when microtrends were useful.



Cons:

--Paper does not provide many actionable steps for using data, or a data-driven approach. Instead, rather broad generalizations are made as to what would be useful (e.g., less manual data entry).

--The implementation section is not informative enough. It would be beneficial to provide more information about the programs used for sorting data, and for moving from one health system to another.

--The figures presented seem disconnected from the text of the paper. Figure 1 should be discussed in the paper, and figure 2 should be expanded to be more descriptive.